# Dosimetric Parameters Related to Acute Radiation Dermatitis of Patients with Nasopharyngeal Carcinoma Treated by Intensity-Modulated Proton Therapy

**DOI:** 10.3390/jpm12071095

**Published:** 2022-06-30

**Authors:** Ko-Chun Fang, Tai-Lin Huang, Kuan-Cho Liao, Tsair-Fwu Lee, Yang-Wei Hsieh, Wen-Ling Tsai, Fu-Min Fang

**Affiliations:** 1Department of Education, Kaohsiung Chang-Gung Memorial Hospital and Chang Gung University College of Medicine, Kaohsiung 83301, Taiwan; erickfang1996@gmail.com; 2Department of Hematology and Oncology, Kaohsiung Chang Gung Memorial Hospital and Chang Gung University College of Medicine, Kaohsiung 83301, Taiwan; victor99@cgmh.org.tw; 3Department of Radiation Oncology, Kaohsiung Chang-Gung Memorial Hospital and Chang Gung University College of Medicine, Kaohsiung 83301, Taiwan; piko@cgmh.org.tw; 4Department of Electric Engineering, National Kaohsiung University of Science and Technology, Kaohsiung 80778, Taiwan; tflee@nkust.edu.tw (T.-F.L.); wewe750422@gmail.com (Y.-W.H.); 5Ph.D. Program in Biomedical Engineering, Kaohsiung Medical University, Kaohsiung 80708, Taiwan; 6Department of Cosmetics and Fashion Styling, Center for Environmental Toxin and Emerging-Contaminant Research, Cheng Shiu University, Kaohsiung 83347, Taiwan; tsai580426@gmail.com; 7Department of Medicine, Chang Gung University College of Medicine, Taoyuan 33302, Taiwan

**Keywords:** nasopharyngeal carcinoma, intensity-modulated proton therapy, acute radiation dermatitis, dosimetric parameters

## Abstract

Background: Growing patients with nasopharyngeal carcinoma (NPC) were treated with intensity-modulated proton therapy (IMPT). However, a high probability of severe acute radiation dermatitis (ARD) was observed. The objective of the study is to investigate the dosimetric parameters related to ARD for NPC patients treated with IMPT. Methods: Sixty-two patients with newly diagnosed NPC were analyzed. The ARD was recorded based on the criteria of Common Terminology Criteria for Adverse Events version 4.0. Logistic regression model was performed to identify the clinical and dosimetric parameters related to ARD. Receiver operating characteristic (ROC) curve analysis and the area under the curve (AUC) were used to evaluate the performance of the models. Results: The maximum ARD grade was 1, 2, and 3 in 27 (43.5%), 26 (42.0%), and 9 (14.5%) of the patients, respectively. Statistically significant differences (*p* < 0.01) in average volume to skin 5 mm with the respective doses were observed in the range 54–62 Cobalt Gray Equivalent (CGE) for grade 2 and 3 versus grade 1 ARD. Smoking habit and N2-N3 status were identified as significant predictors to develop grade 2 and 3 ARD in clinical model, and V58CGE to skin 5 mm as an independent predictor in dosimetric model. After adding the variable of V58CGE to the metric incorporating two parameters of smoking habit and N status, the AUC value of the metric increases from 0.78 (0.66–0.90) to 0.82 (0.72–0.93). The most appropriate cut-off value of V58CGE to skin 5 mm as determined by ROC curve was 5.0 cm^3^, with a predicted probability of 54% to develop grade 2 and 3 ARD. Conclusion: The dosimetric parameter of V58CGE to skin 5 mm < 5.0 cm^3^ could be used as a constraint in treatment planning for NPC patients treated by IMPT.

## 1. Introduction

Radiotherapy with or without the combination of chemotherapy is the major treatment for patients with nasopharyngeal carcinoma (NPC). Proton beam therapy (PBT), with its inherent physical properties of the Bragg peak, has the benefit of dose distribution for cancer treatment. Intensity-modulated proton therapy (IMPT) is a magnetically guided spot scanning PBT, in which all proton spots from complex fields are simultaneously optimized by using an inverse treatment planning system [1]. Growing cancer centers equipped with proton facilities have chosen IMPT to radically treat patients with NPC worldwide. Promising treatment outcomes with the reduction of swallowing-related functional result and the potential increase of survival have been reported compared with the X-ray-based radiotherapy (XRT) [2,3,4,5,6]. However, during the process of modulation of proton beam energy to produce a spread-out Bragg peak to cover the target area, the loss of the skin-sparing effect results in a disadvantage of PBT for the surface area of the skin [7], and a high probability of severe acute radiation dermatitis (ARD), has been observed [2,8].

ARD can progress from erythema to dry desquamation to moist desquamation and even to necrosis. Severe ARD can lead to interruption of radiotherapy course, cause permanent skin changes, diminish aesthetic appeal, reduce quality of life, and potentially negatively influence cancer control [9,10,11]. The occurrence and severity of ARD is not only dependent on the dose and volume of the area irradiated, but also related to the energy or types of radiation sources [12,13]. In clinical practice, multiple factors might contribute to the observed ARD for patients treated with head and neck irradiation, however, the identification of neck skin as a sensitive structure for dose optimization during the process of treatment planning could significantly reduce the skin dose to a tolerable level [12]. Some dosimetric parameters related to severe ARD at chest or extremities have been reported in patients treated by PBT [14,15,16]. As far as we know, a validated dosimetric parameter predictive of ARD for patients treated with IMPT at head and neck area is still lacking. The primary endpoint of the study is to evaluate the dosimetric parameter related to ARD for NPC patients treated with IMPT.

## 2. Materials and Methods

### 2.1. Patient Population

The proton center of Kaohsiung Chang Gung Memorial Hospital in Taiwan started to treat NPC patients in January 2019. Those with newly diagnosed NPC and curatively treated with IMPT for the whole treatment course were recruited. Patients without completion of the proposed treatment course, with protracted course for treatment interruption, or with previous history of radiotherapy at head and neck region were excluded. With the approval of the institutional review board, 62 patients were enrolled for data analysis in the study. The patient characteristics are outlined in Table 1. The median age at the time of diagnosis was 48 (range 31–71) years old. Forty-five (72.6%) patients were male and 19 (30.6%) had the habit of smoking. The distribution of clinical stages based on the American Joint Committee on Cancer (AJCC) 8th edition was stage I, II, III and IVA in 6.5%, 27.4%, 40.3%, and 25.8% of patients, respectively. Fifty-seven (91.9%) patients were treated with IMPT combined with chemotherapy.

### 2.2. Assessment of ARD

ARD was graded using Common Terminology Criteria for Adverse Events version 4.0 (CTCAE v. 4.0) reported weekly by physicians in a prospective fashion at treatment visits (1st to 7th weeks) and 1 week (8th week) and 1 month (11th week) after the completion of IMPT. The grading is grade 1: faint erythema or dry desquamation; grade 2: moderate to brisk erythema, patchy moist desquamation, mostly confined to skin folds and creases, moderate edema; grade 3: moist desquamation in areas other than skin folds and creases, bleeding induced by minor trauma or abrasion; and grade 4: life-threatening consequences, skin necrosis or ulceration of full thickness dermis, spontaneous bleeding from involved site, skin graft indicated.

### 2.3. Technique of IMPT

The technique of IMPT for patients with NPC in the institute was published previously [17]. The technique was delivered by Sumitomo Proton Machine and the treatment planning was carried out by the RayStation treatment planning system (version 7, Raysearch Medical Laboratories, Stockholm, Sweden). Computed tomography (CT) imaging with 1.25 mm per slice was performed for treatment planning purposes with a customized thermoplastic mask for immobilization. Three different dose levels of clinical target volume (CTV) were created. The high dose level of CTV (CTV-H) was defined as the gross tumor and nodes (GTV) with an isotropic extension of 3 mm for the GTV shown in the image studies. The middle dose level of CTV (CTV-M) covered the neighboring risky anatomic structures (e.g., skull base, parapharyngeal space, upper neck lymphatics) of GTV, encompassing micro-metastasis routes of the disease. The low dose level of CTV (CTV-L) included the uninvolved subclinical lymphatics in the lower neck area. The prescribed dose and fractionation for CTV-H, CTV-M, and CTV-L was 69.96 Cobalt Gray Equivalent (CGE), 59.4 CGE, and 52.8–54.0 CGE in 33 fractions, respectively. The organs at risk (OARs) with specified dose constraints were contoured, which included the brain, brainstem, optic nerve, chiasm, lens, cochleas, spinal cord, parotid glands, submandibular glands, oral cavity, constrictor muscle, mandible, larynx, upper esophagus, and thyroid gland. The constrains of these OARs generally followed the guideline recommended [18]. Concerning the OAR of neck skin, skin 5 mm (a layer structure of 5 mm inward from the head and neck contour) was optionally outlined in 41 patients and arbitrarily chosen as a constraint with the request of “as small as possible for V50CGE without compromising the coverage of CTV”. Three- beam directions of posterior, left anterior oblique and right anterior oblique fields with multi-field optimization were typically used for the planning, with the purpose of covering 99.5% of the CTVs and minimizing dose to the OARs. Generally, robust optimization was used to take into consideration of the range (plus 3.5%) and positional uncertainties (plus 3 mm). Robust evaluation, creating 21 plans from the worst- to best-case scenarios, was conducted for the assessment of the planning result. Daily CT-based image guide was performed for set-up accuracy. Adaptive plan was conducted in case of remarkable changes of GTV or patients’ body shape to confirm at least 95% coverage of CTV, and plan sum was used for data analysis in the study.

### 2.4. Chemotherapy

Neoadjuvant chemotherapy with the combination regimens of cisplatin (80 mg/m^2^, day 1) and gemcitabine (1 g/m^2^, day 1 and 8) administered every 3 weeks was given for 3 cycles to those patients with clinical stages III–IVA [19]. Concurrent chemotherapy with intravenous cisplatin 40 mg/m^2^ weekly as a radiation sensitizer was given for 6 to 7 weeks during the treatment course of IMPT for those with clinical stages II–IVA.

### 2.5. Studied Parameters

The studied clinical parameters were the sociodemographic variables (age, gender, smoking habit, and body mass index), comorbidity (hypertension and diabetes mellitus), cancer stage (AJCC stage, T status, and N status) and chemotherapy. The dosimetric parameters that were chosen included the different sizes of CTVs and the volumes (cm^3^) of the skin 5 mm that received 52 CGE (V52CGE), 54 CGE (V54CGE), 56 CGE (V56CGE), 58 CGE (V58CGE), 60 CGE (V60CGE), or 62 CGE (V62CGE), respectively.

### 2.6. Statistical Analysis

The absolute volumes of skin 5 mm with the respective doses for each plan were calculated and average values were assessed for patients who developed different grades of ARD. Volume differences for each specific dose of skin 5 mm were analyzed by two tailed *t*-test in order to be evaluated if there was a statistically significant difference between patients with different ARD grades [20]. Pearson’s chi-squared test was used on the categorical variables and independent t-test was used on the continuous variables to test for differences between patients with grade 1 versus grade 2 and 3 ARD in univariate analysis. Backward stepwise multivariable logistic regression analyses were performed to identify the clinical or dosimetric predictors of grade 2 to 3 ARD, respectively. Logistic plot of the relationship between V58CGE (resulting as the most predictive dosimetric parameter) and the predicted probability of grade 2 and 3 ADR was created. The receiver operating characteristic (ROC) curve was plotted and the area under the curve (AUC) was used to evaluate the performance of the models in prediction of grade 2 and 3 ARD. A ROC curve is a plot of the true positive fraction (sensitivity) versus the false positive fraction (1-specificity). A value of AUC = 1 is a perfect prediction, while a value of 0.5 is equivalent to a random guess. A value of *p* < 0.05 was considered significant. All statistical analysis was processed with IBM SPSS version 22 software.

## 3. Results

### 3.1. Incidence and Severity of ARD

Figure 1 reveals the incidence and severity of ARD observed at the nine time points. During the first 3 weeks, no patients presented remarkable ARD. The maximum ARD grade was 1, 2, and 3 in 27 (43.5%), 26 (42.0%), and 9 (14.5%) of the patients, respectively. No grade 4 ARD was observed. The peak incidence of grade 2 and 3 of ARD was observed during the period of 6th week to 8th week. At 11th week, most grade 2 and 3 ARD had recovered, and 91.9% patients remained with grade 1 or less.

### 3.2. Clinical Predictor of ARD

In univariate analysis of the clinical variables, we observed patients with smoking habit (42.9% versus 14.8%, *p* = 0.011), AJCC stage III-IVA (80.0% versus 48.1%, *p* = 0.011), or N2-N3 status (62.9% versus 29.6%, *p* = 0.007) had a statistically significantly higher probability to present gr 2 and 3 ARD than the counterparts, respectively (Table 2). Entering in the logistic regression model (Table 3), the variables of smoking habit (OR: 5.156, 95% CI: 1.438–18.493, *p* = 0.012) and N2-N3 status (OR: 4.935, 95% CI: 1.583–15.381, *p* = 0.006) were identified as independent predictors of grade 2 and 3 ARD after adjustment of gender, AJCC stage, and body mass index.

### 3.3. Dosimetric Predictor of ARD

The average volume to skin 5 mm with the respective doses referring to patients experiencing grade 1, grade 2, and grade 3 ARD are depicted in Figure 2A. At the dose level of 30–60 Gy, the curves of average volume to skin 5 mm with the respective doses for grades 2 and 3 are nearly identical but relatively higher than grade 1. As shown by the two tailed t-test in Figure 2B, statistically significant differences (*p* values < 0.01) in average volumes to skin 5 mm with the respective doses were observed in the range 54–62 Gy, suggesting that the fraction of dose to skin 5 mm receiving around 1.64–1.88 Gy per fraction is significantly correlated with the severity of ARD. In univariate analysis of the dosimetric parameters, those with higher mean values of V54CGE (29.6 versus 16.3, *p* = 0.009), V56CGE (20.4 versus 9.9, *p* = 0.008), V58CGE (14.0 versus 5.7, *p* = 0.005), V60CGE (9.2 versus 3.2, *p* = 0.008), V62CGE (5.9 versus 1.9, *p* = 0.009), and CTV-H (107.6 versus 67.8, *p* = 0.013) were observed to have a statistically significantly higher probability to present gr 2 and 3 ARD versus grade 1, respectively (Table 2). Entering in the logistic regression model (Table 3), the parameter of V58CGE (OR: 1.139, 95% CI: 1.017–1.276, *p* = 0.024) was identified as the only independent predictor for grade 2 and 3 of ARD after adjustment of the parameters of V54CGE, V56CGE, V60CGE, V62CGE, and CTV-H in the model.

### 3.4. ROC Curve Analysis

The AUC value of V58CGE was 0.74 (0.61–0.87), which was higher than those of the other significant dosimetric parameters in univariate analysis, including V54CGE: 0.68 (0.53–0.82), V56CGE: 0.70 (0.54–0.82), V60CGE: 0.69 (0.55–0.83), V62CGE: 0.68 (0.54–0.82), and CTV-H: 0.66 (0.54–0.81). As shown in Table 4, after adding the variable of V58CGE to the metric incorporating two parameters of smoking habit and N status, the AUC value of the three-parameter metric increases from 0.78 (0.66–0.90) to 0.82 (0.72–0.93). The most appropriate cut-off value of V58CGE to skin 5 mm as determined by ROC curve was 5.0 cm^3^ (Figure 3A). Figure 3B demonstrates the results of the relationship between V58CGE and the predicted probability of grade 2 and 3 ARD (The coefficients of the logistic model β0 and β1 were found to be −0.489 and 0.131 respectively). The predicted probability of grade 2 and 3 ARD corresponding to V58CGE 5.0 cm^3^ was 54%.

## 4. Discussion

As far as we know in the literature review, in the study the dose-volume relationship of skin at head and neck region with ARD for NPC patients treated with IMPT was quantified for the first time. The most important result emerging from the study is that the dose range of 54–62 CGE is clearly associated with the severity of ARD. Despite the relatively small number of patients, the results clearly revealed that the dose-volume parameters of V58CGE to skin 5 mm were highly associated with the risk of developing grade 2 and 3 ARD. Patients with advanced nodal status often receive a higher radiation dose to the neck skin, putting them at a higher risk of severe ARD. However, after adding the variable of V58CGE to the model incorporating the two clinical parameters of smoking habit and N status, the AUC value of the model increases from 0.78 (0.66–0.90) to 0.82 (0.72–0.93), indicating patients with N0-N1 status still have a higher likelihood of grade 2 and 3 ARD when a higher volume of V58CGE at skin surface was created. In terms of the optimal cut-off value potentially to be used as a constraint during planning optimization, limiting V58CGE below approximately 5 cm^3^ (corresponding to 10 cm^2^, slightly more than a 3 × 3 cm^2^ square surface) could keep the risk of grade 2 and 3 ARD less than 54%. The results could be applied during treatment planning, and we could preliminarily predict the risk of grade 2 and 3 ARD for the individual patient and decide accordingly whether to reduce the skin dose.

The dosimetric parameters related to ARD varied in studies with different treatment techniques, the location and thickness of skin surface chosen for dosimetric analysis. The pathophysiology of ARD is usually related to vascularization at dermis. In anatomy, the micro vessels are located between the upper papillary layer and the lower reticular layer of the dermis, which is 1–3 mm thickness from the surface of epidermis. A 2-mm skin contour was used in a study of 70 head and neck cancer patients irradiated by using Helical Tomotherapy. They observed V56 related to skin 2 mm was predictive of grade 2 to 3 ARD with an optimal cut-off value of 7.7 cm^3^ [21]. In consideration of the sensitivity of range uncertainty of PBT and change of body shape during treatment, we used a 5-mm skin contour to minimize the deviation of radiation dose obtained at neck skin. For chest skin, the dosimetric parameters of D10 cm^3^ and V52.5CGE related to skin 5 mm were identified as prognosticators for grade 3 ARD in patients treated with passive-scattering PBT for adjuvant radiotherapy of breast cancer [16]. A comparative study of the skin dose profile of XRT versus passive-scattering PBT for patients with sarcoma revealed V30Gy related to skin 5 mm was predictive of grade 2 and 3 ARD, regardless of the radiotherapy technique [15]. For brain tumor treated by IMPT, V35Gy related to skin 3 mm was reported to be predictive of grade 1 ARD at scalp [22].

In radiation physics, protons have relatively low entrance (skin) doses when monoenergetic beams are used. However, tumor treatment volumes are complex targets with variable thicknesses and depths, requiring modulation of the beam energy to produce a spread-out Bragg peak that covers the target area. This process can result in a significant, and potentially full entrance, dose with loss of the skin-sparing effect characteristic of high-energy XRT, which represents a disadvantage for the surface area of the skin [7]. In clinical settings, however, the comparison of ARD between IMPT and XRT (e.g., IMRT or VMAT) remained inconsistent. After matching between groups, a higher probability of grade 3 ARD was observed in the IMPT group reported by Chou et al. (35% versus 7.5%) [2] and Holliday et al. (40% versus 25%) [13] but not observed by Li et al. (3.6% versus 2.0%) [3].

It should be stressed that the severities of ARD are related to numerous risk factors that have been classified as being patient-related or treatment-related. Patient-related risk factors may include age, gender, smoking, nutritional status, body mass index, comorbidity, or genetic factors. Treatment-related factors include the total radiation dose, the dose fractionation schedule, techniques of radiotherapy, the combination of chemotherapy, and the volume and surface area of irradiated tissue [9,23,24]. For NPC patients, in a large cohort study treated by XRT (IMRT or three-dimensional conformal technique), those treated by IMRT, with lower performance status and receiving multicycle chemotherapy were observed to be predictors of severe ARD [25]. In our patients uniformly treated by IMPT with standardized protocols including total dose and dose per fraction, chemotherapy regimens, skin care, the variables of smoking habit and advanced nodal status were observed to be significant clinical predictors for grade 2 and 3 ARD.

The correlation of smoking with ARD remains inconsistent in the literature for patients treated with XRT [23,26,27]. For PBT, very limited data are available. The association between smoking habit and severity of ARD after PBT has been previously reported in patients with breast cancer [16] but was reported for the first time in patients with NPC in the present study. The mechanism of the effect of smoking on ARD is unknown. However, strong evidence has revealed that smoking adversely impacts the wound-healing process [28]. Tissue hypoxia is viewed as a fundamental mechanism through which smoking disrupts acute wound healing [29]. Smoking impairs the function of several cell types such as neutrophils and macrophages important to inflammatory and bactericidal activity and also compromises oxygen delivery to tissues [30].

Admittedly, there are several limitations to the study. First, the cases were retrospectively reviewed, and the grading of ARD relied on subjective assessments by treating physicians. Second, the use of optically simulated luminescent dosimeters, which would have provided in vivo measurements to correlate with the dosimetric data, is lacking in the studied patients. Third, this study had limited case numbers, which affected the power of predictive models. It should be emphasized that we don’t intent to construct a robust prognostic model ready to be used as clinical criteria in decision-making, but to provide a useful reference for treatment planning to minimize severe ARD. Nevertheless, the individual reaction of skin to proton beam is complex and may be impacted by numerous factors that are difficult to characterize and quantify and further investigation and clinical validation are warranted.

## 5. Conclusions

The study quantified for the first time the relationship between the dosimetric parameters of neck skin (using a 5 mm layer as a surrogate) and the risk of developing ARD in NPC patients treated with IMPT. The ROC curve analysis and AUC values suggest the dosimetric parameter of V58CGE to skin 5 mm < 5.0 cm^3^ could be used as a constraint in treatment planning for NPC patients treated by IMPT. More validation and further prospective investigation are warranted.

## Figures and Tables

**Figure 1 jpm-12-01095-f001:**
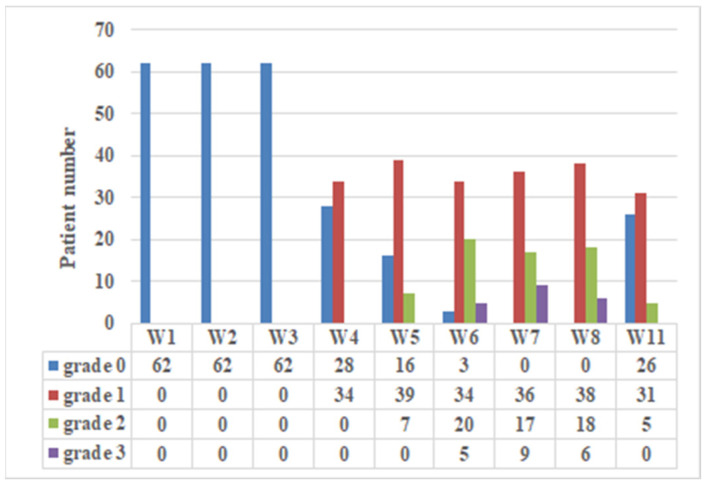
Incidence and severity of acute radiation dermatitis (ARD). The ARD of 62 NPC patients treated with IMPT was graded using CTCAE v. 4.0 reported weekly by physicians in a prospective fashion at treatment visits (1st to 7th weeks) and 1 week (8th week) and 1 month (11th week) after the completion of IMPT.

**Figure 2 jpm-12-01095-f002:**
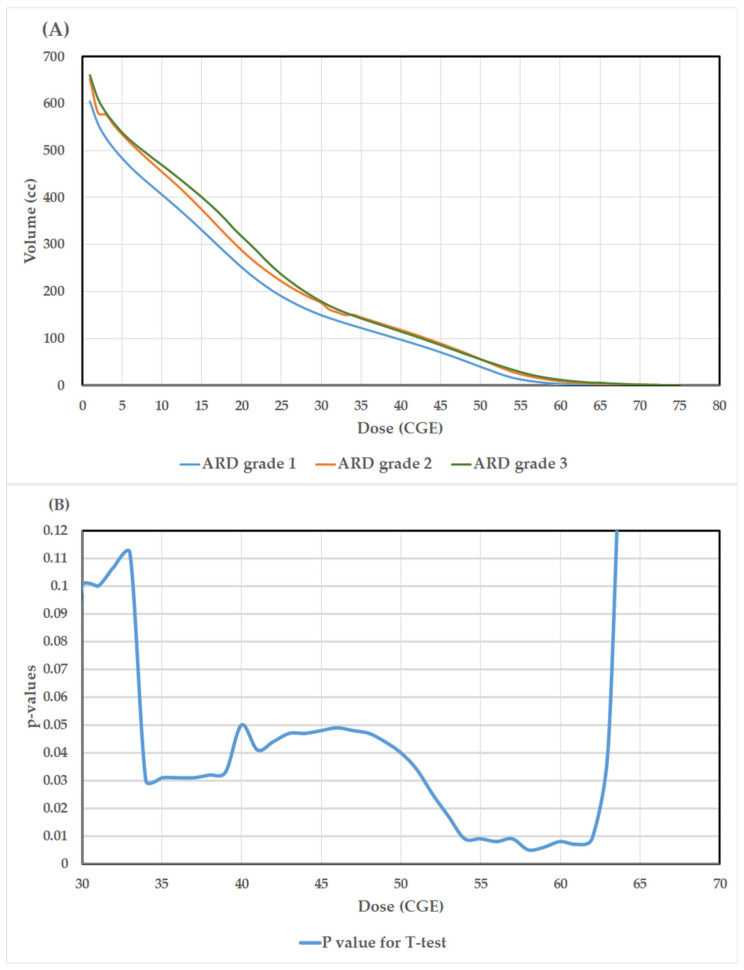
(**A**). The average volumes to skin 5 mm with the respective doses referring to patients experiencing grade 1, grade 2, and grade 3 acute radiation dermatitis (ARD); (**B**). Results of the *p*-values (two-tailed t-test) for the differences of average volume to skin 5 mm with the respective doses between those with grade 1 versus grade 2 and 3 ARD.

**Figure 3 jpm-12-01095-f003:**
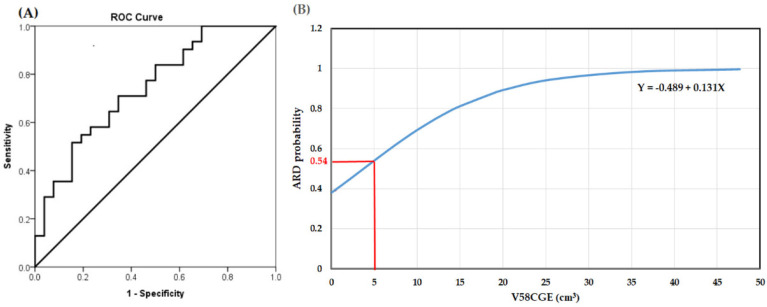
(**A**). Receiver operating characteristic (ROC) curves from the V58CGE model in prediction of grade 2 and 3 acute radiation dermatitis (ARD); (**B**). Logistic plot of the relationship between V58CGE and the probability of developing grade 2 and 3 ARD.

**Table 1 jpm-12-01095-t001:** Patient characteristics (N = 62).

Characteristics	Value	%
Age, median (range), year	48 (31–71)	
Gender, male/female	45/17	72.6/27.4
Smoking habit, yes/no	19/43	30.6/69.4
Body mass index (kg/m^2^), <24.0/≥24.0	25/37	40.3/59.7
Diabetes mellitus, yes/no	6/56	9.7/90.3
Hypertension, yes/no	8/54	12.9/87.1
AJCC stage, I/II/III/IVA	4/17/25/16	6.5/27.4/40.3/25.8
T status, T1/T2/T3/T4	34/9/10/9	54.8/14.5/16.1/14.6
N status, N0/N1/N2/N3	9/23/21/9	14.5/37.1/33.9/14.5
Chemotherapy, yes/no	57/5	91.9/8.1

**Table 2 jpm-12-01095-t002:** Univariate analysis of clinical and dosimetric predictors of grade 2 and 3 acute radiation dermatitis.

Variables	Grade 1 (n = 27)	Grade 2 and 3 (n = 35)	*p* Value
Clinical, No (%)			
Age, ≥48 years	12 (44.4)	18 (51.4)	0.618
Gender, male	17 (63.0)	28 (80.0)	0.160
Smoking habit, yes	4 (14.8)	15 (42.9)	0.011
Body mass index, ≥24.0 kg/m^2^	13 (48.1)	24 (68.6)	0.124
Diabetes mellitus, yes	2 (7.4)	4 (11.4)	0.689
Hypertension, yes	4 (14.8)	4 (11.4)	0.719
AJCC stage, III-IVA	13 (48.1)	28 (80.0)	0.011
T status, T3-T4	7 (25.9)	12 (34.3)	0.583
N status, N2-N3	8 (29.6)	22 (62.9)	0.007
Chemotherapy, yes	24 (88.9)	33 (94.3)	0.689
Dosimetric, mean (SD)			
V54CGE (cm^3^)	16.3 (14.0)	29.6 (22.8)	0.009
V56CGE (cm^3^)	9.9 (9.9)	20.4 (18.5)	0.008
V58CGE (cm^3^)	5.7 (6.8)	14.0 (14.5)	0.005
V60CGE (cm^3^)	3.2 (4.7)	9.2 (11.1)	0.008
V62CGE (cm^3^)	1.9 (3.4)	5.9 (8.8)	0.009
CTV-L (cm^3^)	147.2 (44.8)	164.1 (42.7)	0.151
CTV-M (cm^3^)	187.0 (44.0)	205.9 (71.9)	0.249
CTV-H (cm^3^)	67.8 (40.5)	107.6 (69.9)	0.013

AJCC: American Joint Committee on Cancer staging system 8th edition; CTV-L: clinical target volume, low dose; CTV-M: clinical target volume, middle dose; CTV-H: clinical target volume, high dose; CGE: cobalt gray equivalent.

**Table 3 jpm-12-01095-t003:** Logistic regression model to predict grade 2 and 3 acute radiation dermatitis.

Model	Variable	OR (95% CI)	*p* Value
Clinical model	Smoking habit: yes (ref. no)	5.156 (1.438–18.493)	0.012
	N status: N2-N3 (ref. N0-N1)	4.935 (1.583–15.381)	0.006
Dosimetric model	V58CGE, continuous	1.139 (1.017–1.276)	0.024

Backward stepwise multivariable logistic regression was performed with adjustment of gender, AJCC stage, and body mass index in the clinical model; and adjustment of V54CGE, V56CGE, V60CGE, V62CGE and CTV-H in the dosimetric model; CGE: cobalt gray equivalent; OR: odds ratio; CI: confidence interval; ref: reference.

**Table 4 jpm-12-01095-t004:** AUC value to predict grade 2 and 3 acute radiation dermatitis.

Metric	Parameter	AUC	95% CI
Single parameter	Smoking habit	0.67	0.52–0.81
	N status	0.69	0.55–0.83
	V58CGE	0.74	0.61–0.87
Two-parameter metric	Smoking habit + N status	0.78	0.66–0.90
Three-parameter metric	Smoking habit + N status + V58CGE	0.82	0.72–0.93

AUC: area under the curve, CGE: cobalt gray equivalent, CI confidence interval.

## Data Availability

The data presented in this study are available upon request from the corresponding author. The data are not publicly available due to ethical restrictions.

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
