# Peer review of "Dosimetric Parameters Related to Acute Radiation Dermatitis of Patients with Nasopharyngeal Carcinoma Treated by Intensity-Modulated Proton Therapy"

_jpm, 2022, doi:10.3390/jpm12071095_

Round 1
Reviewer 1 Report
This paper represents a retrospective analysis of the Proton Beam Therapy effects on the skin of patients with nasopharyngeal carcinoma with the purpose of determining possible factors that could increase the probability of Acute Radiation Dermatitis. In addition, the authors suggest an upper dose constrain for skin 5mm. The language is adequate but some parts of the manuscript could be rephrased. The subject under investigation is relevant with the journal’s thematology. Despite having some limitations, this study would be of clinical interest to the readership since this study provides not only clinical but also dosimetric parameters that could be used for this specific cancer case and treatment technique.
Prior to publication, I would like the authors to address the following minor comments:
· Page 1, line 30-31: Please rephrase “ Statistical differences (p<0.01) in dose-volume histogram to skin 5mm 30 were significant in the range 54–62 CGE for grade 2 and 3 versus grade 1 ARD”.
o Statistically significant differences.
o As I understood from the manuscript, the authors used dosimetric parameters taken FROM DVHs of each patient plan in order to perform their analysis. Therefore, the authors did not compare DVHs but dosimetric parameters. A different term should be used instead of DVH.
o The initials of CGE are used for the first time in the abstract, so the authors should write Cobalt Gray Equivalent.
· Page 2, lines 53-54: “However, a high probability of severe acute radiation dermatitis (ARD) was also observed”. A short explanation is required here, not only in Discussion.
· Page 3, line 129: Please change “sum plan” to “ plan sum”.
· Page 4, lines 142-145: “The studied dosimetric parameters included the CTVs (CTV-L, CTV-M, and CTV-H) and the dose-volume histogram (DVH) parameters to skin 5mm, which included volumes (cm3) received 52 CGE...”
Please rephrase as such: “The dosimetric parameters that were chosen included the size of the CTVs and the volumes (cm3) of the skin 5mm that received 52 CGE... “
· Page 4, lines 149-151: Instead of “Absolute DVHs to skin 5mm” please use “The absolute volume of skin 5mm that received the aforementioned doses were taken from DVHs of each patient plan and…”. In addition, the term “DVHs” should be replaced throughout the manuscript, for the aforementioned reason (comment 1). Instead consider rephrasing as such: “Volume differences for each specific dose of skin 5mm were analyzed by two tailed t-test in order to be evaluated if there is statically significance difference between patients with different ARD grades ”.
· Page 5, lines 199-201: “The average skin DVHs referring to patients experiencing grade 1, grade 2, and grade 3 ARD are depicted in Figure 2A. At the dose level of 30-60 curves of skin DVHs for grade 2 and 3 are nearly identical but relatively higher than grade 1.”
It would be more suitable to rephrase as such: “The average volume to skin 5mm with the respective doses referring to patients…”
Author Response
Please see the attachment.
Reviewer 1:
This paper represents a retrospective analysis of the Proton Beam Therapy effects on the skin of patients with nasopharyngeal carcinoma with the purpose of determining possible factors that could increase the probability of Acute Radiation Dermatitis. In addition, the authors suggest an upper dose constrain for skin 5mm. The language is adequate but some parts of the manuscript could be rephrased. The subject under investigation is relevant with the journal’s thematology. Despite having some limitations, this study would be of clinical interest to the readership since this study provides not only clinical but also dosimetric parameters that could be used for this specific cancer case and treatment technique.
Prior to publication, I would like the authors to address the following minor comments:
- Page 1, line 30-31: Please rephrase “ Statistical differences (p<0.01) in dose-volume histogram to skin 5mm were significant in the range 54–62 CGE for grade 2 and 3 versus grade 1 ARD”.
- Statistically significant differences.
- As I understood from the manuscript, the authors used dosimetric parameters taken FROM DVHs of each patient plan in order to perform their analysis. Therefore, the authors did not compare DVHs but dosimetric parameters. A different term should be used instead of DVH.
- The initials of CGE are used for the first time in the abstract, so the authors should write Cobalt Gray Equivalent.
Answer: Thanks for the kindly suggestion. We rephrased “Statistical differences” and “CGE” as suggested in the revised version. We used “average volume to skin 5mm with the respective doses” to represent the content of “DVH” in the manuscript.
- Page 2, lines 53-54: “However, a high probability of severe acute radiation dermatitis (ARD) was also observed”. A short explanation is required here, not only in Discussion.
Answer: Thanks for the kindly suggestion. We added some explanation in the paragraph as: “However, during the process of modulation of proton beam energy to produce a spread-out Bragg peak to cover the target area, the loss of the skin-sparing effect results in a disadvantage of PBT for the surface area of the skin [7], and a high probability of severe acute radiation dermatitis (ARD) was also observed [2,8].”.
- Page 3, line 129: Please change “sum plan” to “ plan sum”.
Answer: Thanks for the kindly suggestion. It was corrected as suggested.
- Page 4, lines 142-145: “The studied dosimetric parameters included the CTVs (CTV-L, CTV-M, and CTV-H) and the dose-volume histogram (DVH) parameters to skin 5mm, which included volumes (cm3) received 52 CGE...”Please rephrase as such: “The dosimetric parameters that were chosen included the size of the CTVs and the volumes (cm3) of the skin 5mm that received 52 CGE... “
Answer: Thanks for the kindly suggestion. We rephrased it as suggested.
- Page 4, lines 149-151: Instead of “Absolute DVHs to skin 5mm” please use “The absolute volume of skin 5mm that received the aforementioned doses were taken from DVHs of each patient plan and…”. In addition, the term “DVHs” should be replaced throughout the manuscript, for the aforementioned reason (comment 1). Instead consider rephrasing as such: “Volume differences for each specific dose of skin 5mm were analyzed by two tailed t-test in order to be evaluated if there is statically significance difference between patients with different ARD grades”.
Answer: Thanks for the kindly suggestion. We rephrased as the followings: “The absolute volumes of skin 5mm with the respective doses for each plan were calculated and average values were assessed for patients who developed different grades of ARD. Volume differences for each specific dose of skin 5mm were analyzed by two tailed t-test in order to be evaluated if there is statistically significant difference between patients with different ARD grades”
- Page 5, lines 199-201: “The average skin DVHs referring to patients experiencing grade 1, grade 2, and grade 3 ARD are depicted in Figure 2A. At the dose level of 30-60 curves of skin DVHs for grade 2 and 3 are nearly identical but relatively higher than grade 1.” It would be more suitable to rephrase as such: “The average volume to skin 5mm with the respective doses referring to patients…”
Answer: Thanks for the kindly suggestion. We rephrased it as suggested.

Reviewer 2 Report
Your study reports on a relevant topic in the treatment of NPC treated by IMPT. Statistics are nicely done.
Reconsider paragraph 2.3, which is a nearly 90% copy of your other paper on this specific topic.
As you sum up, the statistical results you present have to be doubted due to the small number of patients and the non-standardized evaluation of ARD in this retrospective analysis. Nevertheless, the results show hints in a promising direction and are in many aspects according to other results in the literature. In daily routine, though, I would hesitate to use your suggested threshold as a robust parameter.
As you conclude, your hypothesis has to be evaluated in prospective investigations.
